# Efficient Kernel-Based Subsequence Search for Enabling Health Monitoring Services in IoT-Based Home Setting

**DOI:** 10.3390/s19235192

**Published:** 2019-11-27

**Authors:** Antonio Candelieri, Stanislav Fedorov, Enza Messina

**Affiliations:** Department of Computer Science, Systems and Communication, University of Milano-Bicocca, 20126 Milano MI, Italy; fedorov.stas.wrk@gmail.com (S.F.); enza.messina@unimib.it (E.M.)

**Keywords:** data stream analysis, pattern query, kernel learning, dynamic time warping, subsequence search

## Abstract

This paper presents an efficient approach for subsequence search in data streams. The problem consists of identifying coherent repetitions of a given reference time-series, also in the multivariate case, within a longer data stream. The most widely adopted metric to address this problem is Dynamic Time Warping (DTW), but its computational complexity is a well-known issue. In this paper, we present an approach aimed at learning a kernel approximating DTW for efficiently analyzing streaming data collected from wearable sensors, while reducing the burden of DTW computation. Contrary to kernel, DTW allows for comparing two time-series with different length. To enable the use of kernel for comparing two time-series with different length, a feature embedding is required in order to obtain a fixed length vector representation. Each vector component is the DTW between the given time-series and a set of “basis” series, randomly chosen. The approach has been validated on two benchmark datasets and on a real-life application for supporting self-rehabilitation in elderly subjects has been addressed. A comparison with traditional DTW implementations and other state-of-the-art algorithms is provided: results show a slight decrease in accuracy, which is counterbalanced by a significant reduction in computational costs.

## 1. Introduction

Dynamic Time Warping [1] is a technique to find the optimal alignment between two time-series, by considering the possibility to nonlinearly “warp” one time-series by stretching or shrinking it along its time axis. The amount of warping needed for the alignment is then used as a measure of the difference between the two time-series. A typical application of DTW is speech recognition [2], where it is used to determine if two waveforms represent the same spoken phrase. In a speech waveform, the duration of each spoken sound and the interval between sounds can vary, but the overall speech waveforms must have a similar “shape”. In addition to speech recognition, DTW has also been found useful in many other disciplines, including, gesture recognition [3], robotics [4], manufacturing [5], and health monitoring [6,7,8].

Moreover, measuring the similarity between two time-series is a core task for time-series clustering [9], where both data representation and preprocessing are critical choices, as well as the definition of a suitable similarity measure. Recently, in [10], a fuzzy-clustering approach for time-series data has been proposed, where DTW distance is used for comparing pairs of time-series. Other relevant approaches, which could benefit of a more efficient DTW computation, are [11,12].

Indeed, despite its widely adoption in many application domains, a well-known issue of DTW is its computational complexity. Computing DTW between two time-series requires O(NM), where *N* and *M* are the lengths of the two time-series. Therefore, the comparison of a reference time-series with a large data stream (i.e., M≫N), as well as the identification of all the reference repetitions within the data stream, might be computationally very expensive.

The importance of the topic is widely recognized in the scientific community, as highlighted by relevant previous works quoted in Section 2.

The specific contributions of this paper are as follows.
Designing a kernel learning task aimed at approximating DTW to reduce computational burden of the subsequence search.Comparing the proposed kernel-based DTW approximation with traditional DTW-based implementations and other state-of-the-art algorithms.Validating the proposed approach on a simple benchmark toy example (https://www.cs.unm.edu/mueen/FastestSimilaritySearch.html) and on a more complex one, namely, the “User Identification From Walking Activity” dataset, freely downloadable from the UCI Repository.Validating the results through pattern query experiments on a dataset self-rehabilitation dataset, specifically collected in a real-life project. Self-rehabilitation dataset is available from the corresponding author on reasonable request.

The rest of the paper is organized as follows. Section 2 provides the methodological background about DTW, its recent innovations and applications, as well as computational drawbacks in the specific case of subsequence search. Section 3 describes how to learn a kernel to approximate DTW and, consequently, increase the computational efficiency subsequence search within long data streams. Section 4 presents the experimental setting and the datasets used to validate the approach. Section 5 summarizes the experimental results. Finally, the discussion and relevant conclusions are reported in Section 6.

## 2. Backgound

### 2.1. Dynamic Time Warping

The core component of DTW is a data structure named “accumulated cost matrix”, denoted by $D\in R^{N\times M}$, where *N* and *M* are the lengths of the two time-series, $X=(x_{1},\ldots ,x_{N})$, and $Y=(y_{1},\ldots ,y_{M})$ to be compared. Every entry $D_{i,j}$ of this matrix is computed as follows,
(1)$$D_{i,j}=\min\left\{D_{i-1,j-1},D_{i-1,j},D_{i,j-1}\right\}+c\left(x_{i},y_{j}\right)\;\;\forall i=1,\ldots ,N\;\;\;and\;\;\;j=1,\ldots ,M,$$
where $x_{i}$ and $y_{j}$ are the *i*-th and *j*-th values of *X* and *Y*, respectively, and $c:X\times Y\to R_{0}^{+}$ is a cost function. The most widely adopted cost function is the Euclidean distance. The initialization can be simplified by extending the accumulated cost matrix *D* with an additional row and column, specifically $D_{i,0}=\infty$, $D_{0,j}=\infty$, and $D_{0,0}=0$. Then, the recursion in Equation (Equation 1) holds for i=1,…,N and j=1,…,M.

In the general case, the two time-series could be multivariate, and consequently $x_{i}$ and $y_{j}$ could be vectors. Furthermore, the accumulated cost matrix satisfies, by construction, the following identities,
(2)$$D_{i,1}=\sum_{k=1}^{i}c\left(x_{k},y_{1}\right)\;\;\forall i=1,\ldots ,N$$
and
(3)$$D_{1,j}=\sum_{k=1}^{j}c\left(x_{1},y_{k}\right)\;\;\forall j=1,\ldots ,M$$

A warping path is defined as a sequence $p=p_{1},\ldots ,p_{L}$ of positions in *D*, where $p_{l}=\left(i_{l},j_{l}\right)$. The warping path represents an alignment between *X* and *Y*, which assigns the element $x_{i_{l}}$ of the first series to the element $y_{j_{l}}$ of the second series. A warping path must satisfy the following conditions.
Boundary conditions $p_{1}=\left(1,1\right)$ and $p_{L}=\left(N,M\right)$. These conditions enforce the alignment to start and finish at the extremes of the two series, meaning that the first elements of *X* and *Y*, as well as the last ones, must be aligned to each other.Monotonicity condition: $i_{1}\leq i_{2}\leq \ldots \leq i_{L}$ and $j_{1}\leq j_{2}\leq \ldots \leq j_{L}$. This condition simply ensures that if an element in *X* precedes a second one this should also hold for the corresponding elements in *Y*, and vice versa.Step size condition: $p_{l+1}-p_{l}\in \left(1,0),(0,1),(1,1\right)$ for l=1,…,L−1. This condition ensures that no element in *X* and *Y* can be omitted and that there are no replications in the alignment, meaning that all the index pairs contained in a warping path are pairwise distinct. Note that the step size condition implies the monotonicity condition.

The total cost, $c_{p}\left(X,Y\right)$, associated to a warping path *p* between *X* and *Y* is computed as
$$c_{p}\left(X,Y\right)=\sum_{l=1}^{L}c\left(x_{i_{l}},x_{i_{l}}\right)$$

An optimal warping path between *X* and *Y* is a warping path, $p^{*}$, having minimal total cost over all the possible warping paths, as the one one shown in Figure 1. Therefore, the DTW distance between *X* and *Y* is obtained as
(4)$$DTW\left(X,Y\right)=c_{p^{*}}\left(X,Y\right)=\min_{p}c_{p}\left(X,Y\right)$$

Note that DTW is symmetric if the cost function c(·,·) is symmetric. However, DTW is generally not positive definite and does not always satisfy the triangle inequality. The following figure provides an example of accumulated cost matrix and the associated optimal warping path. The closer the optimal warping path to the diagonal, the lower the misalignment between the two series.

The optimization problem (Equation 4) is solved using dynamic programming, with complexity O(NM). The optimal warping path algorithm is summarized in the following Algorithm 1.
**Algorithm 1** Optimal warping path algorithm**Input** Accumulated cost matrix *D*
**Output** optimal warping path p*1:i←N,j←M,p1←(i,j),l←12:**while**i≠1 OR j≠1
**do**3:    l←l+14:    **if**
i=1
**then**
5:        $p_{l}\leftarrow \left(i,j-1\right)$6:        j←j−17:    **else**8:        **if**
j=1
**then**
9:           $p_{l}\leftarrow \left(i-1,j\right)$10:           i←i−111:        **else**12:           $d^{*}\leftarrow \min\left\{D_{i-1,j-1},D_{i-1,j},D_{i,j-1}\right\}$13:           **if**
$d^{*}=D_{i-1,j}$ OR $d^{*}=D_{i-1,j-1}$
**then**
14:               i←i−115:           **end if**16:           **if**
$d^{*}=D_{i,j-1}$ OR $d^{*}=D_{i-1,j-1}$
**then**
17:               j←j−118:           **end if**19:           $p_{l}\leftarrow \left(i,j\right)$20:        **end if**21:    **end if**22:**end while**23:$p^{*}=reverse\left(p\right)$


The reverse operation at the end of the algorithm is necessary because the length *L* of the optimal warping path $p^{*}$ is unknown a priori. Indeed, the optimal warping path is computed, according to the dynamic programming paradigm, in a reverse order starting from the position (N,M) to the position (1,1). Therefore, the reverse operation allows to give, as output, the optimal warping path, represented as a sequence of positions coherent with the initial definition.

A commonly adopted DTW variant is to impose global constraint conditions on the admissible warping paths with the aim to prevent undesired alignments by controlling the route of a warping path. Two widely adopted global constraints are the Sakoe–Chiba band [13] and the Itakura parallelogram [14]. Besides the prevention of undesired alignments, global constraints can also speed up DTW computation, because they in effect limit the length, *L*, of $p^{*}$.

A review of the research efforts in optimizing both the efficiency and effectiveness of DTW-based algorithms for similarity search, clustering, and classification is presented in [15]. Here, different variants of DTW are discussed, such as constrained DTW, multidimensional DTW, asynchronous DTW, along with optimization techniques for improving DTW efficiency, such as lower bounding, early abandoning, run-length encoding, bounded approximation, and hardware optimization. Some relevant aspects of DTW optimization are presented in [16], where an example of approximation of the accumulated cost matrix *D* is given, and in [17], where the distance calculations for univariate DTW are accelerated. In [18], a exotic approach to increase robustness of similarity measure by constructing a matrix over the derivative approximation of neighborhood samples can be found, namely, Derivative DTW. Finally, with respect to the topic of DTW approximation via kernel, the authors of [19] propose a kernel aimed at learning the principal global alignments for the given data by using the hidden structure of the alignments from the training data. This approach is presented as more computationally efficient when compared to previous kernels on DTW distance, such as GA kernel [20,21] and Gaussian DTW kernel [22].

### 2.2. Dynamic Time Warping for Subsequence Search

The problem of subsequence search, based on DTW, is described in [1], and is also known as subsequence DTW or DTW-based subsequence search. In this task, the time-series to be compared are characterized by significant difference in their lengths, i.e., M≫N, where the shortest one is called reference pattern, that is, a specific sequence to be searched for within the second longer time-series. Indeed, instead of searching for a global alignment between the two series, the goal is to find at least one subsequence of the reference pattern within the longer data stream, with optimal fitting (i.e., minimal DTW). Let us denote by $a^{*}$ and $b^{*}$ the indices representing the beginning and end of the subsequence within *Y*, with $1\leq a^{*}\leq b^{*}\leq M$. These indices are identified by solving the following optimization problem,
(5)$$\left(a^{*},b^{*}\right)=\underset{\left(a,b\right):1\leq a^{*}\leq b^{*}\leq M}{arg min}DTW\left(X,Y_{a:b}\right),$$
where $Y_{a:b}$ is the subsequence $\left(y_{a},\ldots ,y_{b}\right)$ in *Y*.

Optimization problem (Equation 5) can be solved by applying a modification in the initialization of the previous DTW algorithm, consisting of replacing (Equation 3) with
$$D_{1,j}=c\left(x_{1},y_{j}\right)$$

In other words, contrary to the identities in Equation (Equation 3), the starting position of the subsequence $a^{*}$ does not provide any value, except its own cost, and therefore the cost of positioning $b^{*}$ depends only on the DTW between the reference pattern and the chosen subsequence. The remaining values of the accumulated cost matrix *D* are defined as in the basic DTW algorithm.

The index $b^{*}$ is determined as $b^{*}={arg min}_{b=1,\ldots ,M}D_{N,b}$. In case $b^{*}$ is not unique, the lexicographic order can be used to select among the multiple choices. Given the value $b^{*}$, then $a^{*}$ is obtained by applying the optimal warping path algorithm, starting from position $\left(N,b^{*}\right)$. Finally, the resulting optimal warping path $p^{*}=\left(p_{1},\ldots ,p_{L}\right)$ must be reduced to $\left(p_{l},\ldots ,p_{L}\right)$, where $p_{l}$ is the maximum index such that $p_{l}=\left(a^{*},1\right)$, with l∈{1,…,L}. Therefore, the optimal warping path between *X* and $Y_{a^{*}:b^{*}}$ is given by $\left(p_{l},\ldots ,p_{L}\right)$, and, roughly speaking, all the elements preceding $y_{a^{*}}$ and those following $y_{b^{*}}$ are not considered in the alignment and, consequently, do not account for additional costs to DTW.

In the following, we summarize how the subsequence search algorithm can be extended to find multiple repetitions of the reference pattern *X* within the longer data stream *Y*. First, we introduce, as reported in [1], the distance function Δ:[1:M]→R, with $\Delta \left(b\right)=D_{N,b}\;\;b=1,\ldots ,M$, which assigns the minimal DTW that can be computed between the reference pattern *X* and a subsequence in *Y* ending in $y_{b}$. Given *b*, the starting index $y_{a}$ of the searched subsequence is identified through the optimal warping path algorithm revised for subsequence search. The procedure is summarized in Algorithm 2.

Step 8 of the Algorithm 2 is particularly important. As the elements of the accumulated cost matrix are computed accordingly to Equation (Equation 1), we set Δ(b)=∞ in the neighborhood of the optimal value $b^{*}$ to avoid subsequences already found (optimization process in step 4) as well as pathological cases of a very short time-series in its neighborhood.

To guarantee no false dismissals in similarity query processing and efficiently prune a significant number of the search candidates, leading to a reduction in the search cost, the authors of [23] propose a fast similarity search method (FTW).
**Algorithm 2** DTW-based subsequence search algorithm**Input**
reference pattern X=(x1,…,xN), a longer data stream Y=(y1,…,yM), with M≫N, and a threshold τ**Output** a list L of repetitions of *X* within *Y* having, individually, a DTW lower than τ. The list is ranked depending on the individual DTW1:L=∅2:compute the accumulated cost matrix *D* between *X* and *Y*
3:compute the distance function Δ
4:find $b^{*}={arg min}_{b\in \{1,\ldots ,M\}}\Delta \left(b\right)$
5:**If**Δ(b)>τ**then** STOP 6:find $a^{*}$ by using **Optimal warping path algorithm** but initialized in j=b instead of j=M
7:updating L as: $L=L\cup Y_{a^{*}:b^{*}}$
8:Set Δ(b)=∞ for every *b* in a suitable neighborhood of $b^{*}$
9:GO TO STEP 3


## 3. Learning a Kernel to Approximate DTW

### 3.1. Time-Series Kernels via Alignments

A number of global alignment kernels have been proposed in literature with the aim to extend DTW to a kernel-based estimation method. The underlying idea is to avoid the problem of searching exactly the optimal warping path while learning a kernel approximating the DTW value between two time-series. Kernel methods have shown to be promising for learning complex models by implicitly transforming a simple representation, like mapping typical Euclidean distance into a high-dimension feature space [24]. The main obstacles for applying usual kernel methods to time-series are due to two distinctive characteristics of time-series: (a) variable length and (b) dynamic time scaling and shifts. Furthermore, direct use of DTW leads to a not positive definite kernel that does not provide a convex optimization problem [20]. To overcome these obstacles, a family of global alignment kernels have been proposed by taking soft-max over all possible alignments in DTW to give a positive definite kernel [20,21,25]. However, the effectiveness of the global alignment kernels is impaired by the diagonal dominance of the resulting kernel matrix proportional to the difference in the size between the two time series [21], which is the case of subsequence search. In [26], a random features mapping method for time-series embedding is proposed: the idea is to use an explicit mapping to represent any time-series through its alignments to a set of randomly chosen “basis times-series”, having a small length. This significantly reduces computational cost. Starting from similar considerations, our approach aims at learning a kernel, based on random features mapping, to use for efficiently solving the subsequence search problem.

### 3.2. Learning a Kernel for Subsequence Search

Consider two multimodal time-series $X=(x_{1},\ldots ,x_{N})$ and $Y=(y_{1},\ldots ,y_{M})$ with M≫N, where *X* represents the reference pattern to be searched in *Y*. Let us now randomly generate a sequence $S=\{s_{1},s_{2},\ldots ,s_{R}\}$ of *R* basis time-series, where $s_{i}\in R^{d\times L_{i}}\;\;\forall i=1,\ldots ,R$, *d*-dimension of data and $L_{i}\in \left[L_{\min},L_{\max}\right]$ is the length of the *i*-th basis time-series $s_{i}$ (usually $L_{i}\ll M$ and $L_{i}<N$), and where $L_{\min}$ and $L_{\max}$ are the minimum and maximum length allowed. *R*, $L_{\min}$, and $L_{\max}$ are technical parameters of the algorithm and must be tuned experimentally. The considerations on computational complexity given in the following provide useful relations for setting up the values of these parameters suitably.

According to the authors of [26], if the set *S* of basis time-series is sampled from a normal distribution, it shows good performance in further construction of kernel.

Let us define the feature map $\Phi_{S}\left(X\right)={\left(\phi_{s_{1}}\left(X\right),\ldots ,\phi_{s_{R}}\left(X\right)\right)}^{T}$, where the *i*-th component of $\Phi_{S}\left(X\right)$ is the alignment between *X* and the random series $s_{i}$. We consider DTW as measure of this alignment, thus we must compute the accumulated cost matrix *D* for every entry of the feature vector:
(6)$$\Phi_{S}\left(X\right)={\left(DTW\left(X,s_{1}\right),\ldots ,DTW\left(X,s_{R}\right)\right)}^{T}$$

The mapping (Equation 6) provides an *R* dimensional vector without correspondence to dimensionality of original time series, therefore it is used in the further construction of a kernel able to work with time-series having, originally, different length. Although DTW must be computed *R* times, by keeping $RL_{\max}<N$ the computational cost is low due to the reduced length of each $s_{i}$, more specifically. Indeed, the cost for computing $\Phi_{s_{i}}\left(X\right)$, in the worst case, is $O(NRL_{\max})$. Moreover, parallel computing can be used to further improve efficiency, since the computation of each component of the vector $\phi_{S}\left(X\right)$ is embarrassingly parallel, as depicted in the following figure, where we assume, for sake of simplicity, to have *R* different processors available.

Given the two time-series *X* and *Y*, $\Phi_{S}\left(X\right)$ and $\Phi_{S}\left(Y\right)$ are their associated representation in the space spanned by their DTW with respect to the set of basis series *S*. Note that whichever is the length of *X* and *Y*, their mappings, $\Phi_{S}\left(X\right)$ and $\Phi_{S}\left(Y\right)$, have the same length, that is, *R*. Contrary to the authors of [26], we decide to preserve the same set of basis series, *S*, that lead us to equality of spanned spaces and let us fairly compare time-series regardless of the diagonal dominance problem given by skewed data. To introduce the positive definite distance, we decided to use a nonlinear kernel, i.e., Radial Basis Function (RBF) kernel, comparing $\Phi_{S}\left(X\right)$ and $\Phi_{S}\left(Y\right)$:
$$K\left(\Phi_{S}\left(X\right),\Phi_{S}\left(Y\right)\right)=\exp(-\frac{1}{2\gamma^{2}}∥\Phi_{S}\left(X\right)-\Phi_{S}\left(Y\right){∥}^{2})$$
where ∥·∥ denotes Euclidean norm and γ is the length-scale parameter. Although a simple linear kernel could be adopted, such as in [26], this reduces the approximating capability of the approach. On the other hand, the adoption of the RBF kernel leads to the need for optimizing the length-scale hyperparameter.

As kernel measures the similarity between the two time-series, we used the following formula to define our DTW kernel-based distance,
$$d_{K}\left(X,Y\right)=1-K\left(\Phi_{S}\left(X\right),\Phi_{S}\left(Y\right)\right)$$

The computational complexity of $d_{K}\left(X,Y\right)$ is given by the computational complexity of $\Phi_{S}\left(X\right)$ and $\Phi_{S}\left(Y\right)$, leading to $O(RL_{max}N+RL_{max}M)$. Let us denote $L_{max}=\alpha N$, with α∈(0,1); then, to be more efficient than traditional DTW, the following relation must be satisfied,
(7)$$O\left(RL_{max}N+RL_{max}M\right)=\rho O\left(NM\right),$$
with ρ∈(0,1). From Equation (Equation 7) it is possible to derive the relation linking the technical parameters of the proposed approach. More precisely,
$$RL_{max}N+RL_{max}M=\rho NM$$
and consequently
(8)$$R=\frac{\rho M}{\alpha (N+M)}$$
where ⌊R⌋ represents the maximum cardinality of *S*, given the value of ρ and α.

Now, two different cases are to be considered: if N≃M, i.e., using the kernel-based DTW approximation to compare two series with similar length, then R=ρ/2α; whereas if N≪M, i.e., searching for subsequence in a longer data stream, then R≃ρ/α. It is interesting to note that, in both two cases, the value of *R* does not depend on the lengths *N* and *M*. Let us consider a simple example where α=0.1 and ρ=0.5, meaning that we want a reduction of computational cost of 50% with respect to DTW and to use a basis series no longer than 10% of the reference time-series *X* ($L_{max}=\alpha N$). Then, according to Equation (Equation 8), we obtain R=5. This value might be too small to obtain a good DTW approximation. However, thanks to the proposed parallel computation schema, it is possible to increase the number of basis series up to $\bar{R}=n_{pp}\left\lfloor R\right\rfloor$, with $n_{pp}$ being the number of parallel processes. For instance, with $n_{pp}=6$, parallel processors can use the $\bar{R}=30$ basis time-series, which can be considered statistically significant.

With respect to the subsequence search problem, it is now possible to replace DTW in Equation (Equation 5) with its kernel-based approximation:
(9)$$\begin{array}{c} \left(a^{*},b^{*}\right)=\underset{(a,b):1\leq a<b\leq M}{arg min}d_{K}\left(X,Y_{a:b}\right)\\ \mathrm{subject}\;\mathrm{to}:\\ b-a\leq \beta^{+}N\\ b-a\geq \beta^{-}N \end{array}$$
where $\beta^{+}$ and $\beta^{-}$ are two coefficients to set up, representing, respectively, the largest and the smallest length of the possible subsequence with respect to *N*, that is, the length of the reference time-series *X*. Values of these coefficients depend on the specific application: suitable ranges are $\beta^{+}\in \left[1,2\right)$ and $\beta^{-}\in \left(0,1\right]$.

Due to the nature of $d_{K}\left(X,Y\right)$, which requires the computation of $\Phi (Y_{a:b})$ for each pair (a,b), Equation (Equation 9) is a Black-Box Optimization (BBO) problem. We solve it via Bayesian Optimization (BO) [27]. BO is a technique successfully applied for automating the configuration of Machine Learning algorithms, such as autoML [28,29], as well as complex Machine Learning pipelines [8]. The aim is to obtain a good solution of (Equation 9) by trying a limited number of possible pairs (a,b), i.e., function evaluations.

Solving Equation (Equation 9) via BO requires $O\left(\alpha RN^{2}+\alpha RN\beta^{+}N\eta \right)+O_{BO}$, where $\beta^{+}N$ is the maximum length of the subsequence $Y_{a:b}$, η is the maximum number of function evaluations, and the term $O_{BO}$ summarizes the computational cost of BO. This cost is usually dominated by $O(t^{3})$ in the case of Gaussian processes-based BO, where *t* is the number of function evaluations performed (t=1,…η), so it increases with the number of pairs (a,b) evaluated. In any case, η can be chosen to be η≪N.

To be more efficient than DTW-based subsequence search, the following relation must be satisfied,
(10)$$O\left(\alpha RN^{2}+\alpha RN\beta^{+}N\eta \right)+O_{BO}<O\left(MN\right)+O_{owp}$$
where $O_{owp}$ summarizes the complexity of the optimal warping path algorithm (Algorithm 1). For this analysis, we can consider negligible both $O_{BO}$ and $O_{owp}$. Therefore,
(11)$$\alpha RN^{2}+\alpha RN\beta^{+}N\eta <MN$$

One can now set
(12)$$\alpha RN^{2}+\alpha RN\beta^{+}N\eta =\bar{\rho }MN$$
with $\bar{\rho }\in \left(0,1\right)$, and consequently
(13)$$\eta =\left\lfloor \frac{\bar{\rho }M-\alpha RN}{\alpha RN\beta^{+}}\right\rfloor$$

For example, consider two time-series *X* and *Y* having length N=100 and M=10000, respectively. If α=0.1 and ρ=0.5, it follows from Equation (Equation 8) that R=5. By further setting $\bar{\rho }=0.5$ and $\beta^{+}=1.5$, it follows from (Equation 13) that η=66, providing an upper bound on the number of function evaluations to perform during BO.

### 3.3. Extension to Multiple Reference Patterns

This section summarizes how we can extend our approach to implement a subsequence search when multiple reference patterns, $X_{i}$, with i=1,…,n, have to be searched for within multiple data streams $Y_{j}$, with j=1,…,m. The first step involves generating a specific kernel for each reference pattern by learning the best value of the length-scale $\gamma_{i}$ directly from the data:(14)$$\gamma^{*}=\underset{\gamma \in R^{n}}{arg min}\frac{2}{n(n-1)}\sum_{i=1}^{n-1}\sum_{j=i+1}^{n}\left|\left(1-K_{\gamma_{i}}\left(\phi_{S}\left(X_{i}\right),\phi_{S}\left(X_{j}\right)\right)-DTW\left(X_{i},X_{j}\right)\right)\right|$$
where $\gamma ={\left(\gamma_{1},\ldots ,\gamma_{n}\right)}^{T}$ is the vector of the length-scale values of the kernels associated to the corresponding reference patterns $X_{i},i=1,\ldots ,n$. The idea is to minimize, for each possible pairs of time series $X_{i}$ and $X_{j}$, the difference between the DTW and its kernel approximation computed on the mapping of the two time series, that is, $\phi_{S}\left(X_{i}\right)$ and $\phi_{S}\left(X_{j}\right)$. Given that the matrix is symmetric by construction, problem (Equation 14) can be reduced to minimize the upper triangular matrix of the errors between each pair of reference patterns.

As the range of RBF kernel is [0,1], DTW is preliminary rescaled in the same interval, as reported in step 8 of the following algorithm. Algorithm 3’s parameters have to be tuned manually, according to the procedure previously described.
**Algorithm 3** Learning a kernel for approximating DTW in the case of multiple references and multiple data streams**Input***n* reference patterns {Xi}i=1…n, *m* data streams {Yl}l=1…m and parameters R,Lmin,Lmax,β−,β+,σ2.**Output** a matrix K∈Rn×m containing the kernel values.1:generate a set of *R* “basis” time series S={s1,s2,…,sR}, where each si∼N(0,σ2)
2:**for**i=1:n**do**3:    compute ΦS(Xi)=(DTW(Xi,s1),…,DTW(Xi,sR))T
4:    **for**
j=1:n
**do**
5:        $M_{i,j}=DTW\left(X_{i},X_{j}\right)$
6:    **end for**7:**end for**8:normalize entries of the matrix M
9:choose $\gamma^{*}={arg min}_{\gamma \in R^{n}}\frac{2}{n(n-1)}\sum_{i=1}^{n-1}\sum_{j=i+1}^{n}\left|\left(1-K_{\gamma_{i}}\left(\phi_{S}\left(X_{i}\right),\phi_{S}\left(X_{j}\right)\right)-M_{i,j}\right)\right|$
10:compute every entry of K as $K_{i,l}=d_{K,\gamma^{*}}\left(X_{i},Y_{l,a^{*}:b^{*}}\right)$, where $a^{l},b^{*}$ are obtained solving (Equation 9).


## 4. Experimental Setting

### 4.1. Organization of the Experiments

To validate the proposed kernel-based DTW approximation, we have considered three different experimental settings with different levels of complexity. More in detail, the first experiment considers the simplest case, that is, a univariate setting, where a given reference pattern is searched for within a longer data stream. The second experiment extends the analysis to a multivariate setting with multiple reference patterns. Finally, the third experiment refers to a real-life application (i.e., self-rahabilitation at home).

### 4.2. Experiment 1: A Univariate Case

The first experiment considers a univariate benchmark dataset consisting of accelerometer data collected on a Sony AIBO robot dog. The reference pattern to be searched (consisting of 100 data points) is related to acceleration collected when the dog was walking on a carpet, whereas the longer data stream refers to a sequence of data collected in three different conditions while robot was walking on cement (for 5000 data points), on carpet (for 3000 data points), and again on cement (for 5000 data points). Both the reference patterns and the data stream can be downloaded for free from https://www.cs.unm.edu/mueen/FastestSimilaritySearch.html. The goal is to search for the first 10 and 25 best-matching repetitions of the reference pattern into the data stream. Our algorithm was compared with three other approaches:MASS [30]: a fast similarity search algorithm for subsequences under Euclidean distance and correlation coefficient (experiments refer to MASS under Euclidean distance, only). A strong assumption of MASS is that the identified subsequences have the same length of the reference.DTW with fixed window: based on the same assumption of MASS but using DTW instead of Euclidean distance.DTW-based subsequence search algorithm described in Algorithm 2.DTW-based Kernel constructed to approximate the exact DTW.

### 4.3. Experiment 2: A Multivariate Case

The dataset we considered for this experiment is a benchmark dataset for user identification from walking activity [31], which can be freely downloaded from the UCI Repository website: (https://archive.ics.uci.edu/ml/datasets/User+Identification+From+Walking+Activity). The dataset refers to accelerometer data (i.e., acceleration on the *x*, *y*, and *z* axes) acquired through an Android smartphone positioned in the chest pocket and from 22 participants walking in the wild over a predefined path. Data information:Sampling frequency of the accelerometer: DELAY_FASTEST with network connections disabled.A separate file for each participant.Every row in each file consists of time-step, *x* acceleration, *y* acceleration, and *z* acceleration.

From the 22 data streams—one for each participant—a small portion of data (200 data points, that is, ~6 s) is extracted and considered as reference pattern for the corresponding user. The size of the reference pattern has been selected after some preliminary exploratory analysis on the entire set of recordings. Given a data stream, the goal is to associate it to the corresponding user. To do this, the data stream is associated to the user whose reference pattern results in the highest number of best matching repetitions. This the experiment is specifically devoted to test the Algorithm 3.

### 4.4. Experiment 3: A Real-Life Application

This dataset was specifically collected for designing and developing a digital service for supporting self-rehabilitation at home. We used three Inertial Measurement Units (IMUs) worn over the chest, the wrist (of the dominant arm), and the ankle (of the dominant leg), respectively. The sensors permit to acquire several measures over the three axes (i.e., orientation, acceleration, and velocity) with a frequency of 10 Hertz. For the purpose of this study, we considered the acceleration measures only, as they are the most relevant information about the movement performed by the subject. Data refers to an over 60-year-old woman performing the following schedule of five exercises.
Flexo-extension of the knee (sit-down position)Raise and lower the arms (sit-down position)Rotate the torso (sit-down position)Back extension of the legs (stand-up position)Light squat (stand-up position)

These rehabilitation exercises were initially performed by the subject in the clinical setting under the supervision of a qualified trainer: the resulting reference patterns, certified by the trainer, are assumed as gold standards. Then, the subject performed a self-rehabilitation session at home and acceleration data were collected from the wearable sensors. Within the collected data stream, we searched for each one of the five reference patterns.

The correct identification of reference patterns was validated via visual inspection. Note that the exercises schedule (i.e., order and number of repetitions) was planned before the self-rehabilitation session, making it easy for the clinician to identify which part of the data stream corresponds to a specific exercise. The repetitions identified by the algorithms—traditional DTW and the kernel-based approximation—which are in the correct portion of data were considered as hit; otherwise, miss.

### 4.5. Computational Setting

All the experiments have been performed on an Intel Core i7-7700HQ CPU at 2.80 GHz, 16 GB RAM, Windows 10 OS (64 bit). Software used: R (3.6.0) and Matlab (R2019a).

As we are interested in comparing our approach with traditional DTW, which is not parallel, we do not exploit the parallelization schema reported in Section 3.2, Figure 2. Thus, time reduction reported in the results could be even more relevant if parallelization is used.

## 5. Results

### 5.1. Results of Experiment 1

Although all the algorithms correctly identified the first 10 best-matching repetitions (Figure 3) within the portion of data collected while the robot dog was walking on the carpet, there remain some slight differences between the DTW-based Subsequence Search and MASS algorithm. Both the DTW-based subsequence search (i.e., Algorithm 2) and the proposed kernel-based approach allow temporal deformation or subsequences with a different length (i.e., the subsequence can be faster or slower, still maintaining the same shape of the reference pattern). Finally, the proposed kernel-based DTW approximation reduces the computational time more than 2 times (3.4 s compared to 8.1 s of Algorithm 2).

Of greater interest, when the number of repetitions to be found is increased from 10 to 25, the results provided by the four approaches are significantly different, as depicted in Figure 4. MASS and the DTW with fixed window size have identified, respectively, 9 and 6 subsequences outside the portion of data collected while the AIBO robot was walking on carpet. On the contrary, only four out of 25 subsequences identified by the DTW-based Subsequence Search (Algorithm 2) are outside the correct portion of data, and thus more effectively manage the temporal deformation between the subsequence and reference pattern by this algorithm, generate a more accurate identification. The proposed kernel-based DTW approximation results in a higher number of errors than DTW-based subsequence search (i.e., 8 subsequences outside the correct portion of data), that is, lower than MASS and comparable to DTW with fixed windows. Errors are due to the approximating nature of the approach but are counterbalanced by the significant reduction in computational time.

### 5.2. Results on Experiment 2

According to Experiment 1, the DTW-based subsequence search and kernel-based DTW approximations were more effective in identifying those subsequences, which are similar in shape to the reference. Therefore, we have decided to focus Experiment 2 on these approaches. The main differences with Experiment 1 are that, in this case, data streams are multivariate and 22 different references and data streams are considered. Thus, this experiment is devoted to validate the extension of the proposed approach to the multi-references case, summarized in Algorithm 3. The task consists in identifying, for each pair “reference–data stream”, the number of matching subsequences whose distance from the specific reference is lower than a given threshold. We named these repetitions “coherent repetitions”. Algorithm 2 was applied in two different ways: (a) directly on the multivariate data and (b) on each dimension of the data streams, separately (univariate approach). The sum of the coherent repetitions over the dimensions provides a measure on the quality of the matching between the given reference and data stream. The kernel-based DTW approximation algorithm was applied on the multivariate data streams, only. Figure 5 summarizes the confidence levels between each pair of reference pattern and data stream, respectively for DTW-based sequence search (on the left) and kernel-based DTW approximation (on the right), where each row of the matrix is associated to a reference and each column is associated to the data stream where the reference belongs to. The confidence level is computed as the number of repetitions of the reference within the data stream, in the case of DTW-based subsequence search, and values of the kernel-based distance in the case of the proposed approach. As both represent the same measure of confidence level, we simply rescaled it into [0,1]; more precisely, in the case of the DTW-based subsequence search, from every entry of the matrix, the minimum value on the corresponding row was subtracted, and then the result were divided by the difference between maximum and minimum values on that row. As the kernel-based DTW approximation is by definition in [0,1], it does not require any further rescaling.

The DTW-based sequence search was able to correctly associated 19 out of 22 users to their corresponding data streams, whereas our kernel-based DTW approximation algorithm correctly associated 17 out of 22. The proposed kernel-based DTW approximation resulted in a higher error due to its approximating nature; however, this higher error was counterbalanced by a significant reduction in computational time, which is in percentage similar to the previous experiment: 927 s compared to 2056 s required by Algorithm 2. The reduction of the computational cost is slightly lower than the previous case due to the further effort required to optimize the value of the kernel’s hyperparameter via Bayesian Optimization.

### 5.3. Results of Experiment 3

Recall that the correct identification of the five reference patterns was validated by the clinician via visual inspection of the data stream collected during a self-rehabilitation session performed by the subject. Table 1 reports the results for this experiment. In particular, the subsequences identified in the correct portion of the data stream (i.e., “hit”) are labeled by “Yes”, whereas the “miss” ones are labeled by “No”. A label “-” denotes no subsequences identified.

From Table 1, it is possible to notice that, except for the second rehabilitation exercise, the traditional DTW-based subsequence search and the proposed kernel-based DTW approximation are quite aligned. This could be due to the basis time-series randomly chosen, which might be not able to provide a good approximation of this specific reference pattern. Even in this case, the reduction in terms of computational costs is similar to previous experiments: ~21 s compared to 37 s required by Algorithm 2.

## 6. Conclusions

We present an efficient approach of the subsequence search problem in data stream where DTW computation is approximated through kernel learning in the space induced by a feature embedding derived on a set of randomly generated basis time-series.

The validation on three different “small-scale” case studies have been highlighted the potential advantages offered by the proposed approach; basically, a good approximation and a significant lower computational time with respect to a traditional DTW-based implementation. The parallelization schema proposed in Figure 2 makes this approach also applicable to large-scale data streams.

A relevant limitation of the approach is its strong dependence on the initial set of randomly generated basis time-series. Future work should be devoted to define a suitable set of these time-series leading to a robust and effective embedding, starting from the findings provided in [32]. Finally, the relation between the embedding and the (control of) approximation error should be more deeply investigated.

## Figures and Tables

**Figure 1 sensors-19-05192-f001:**
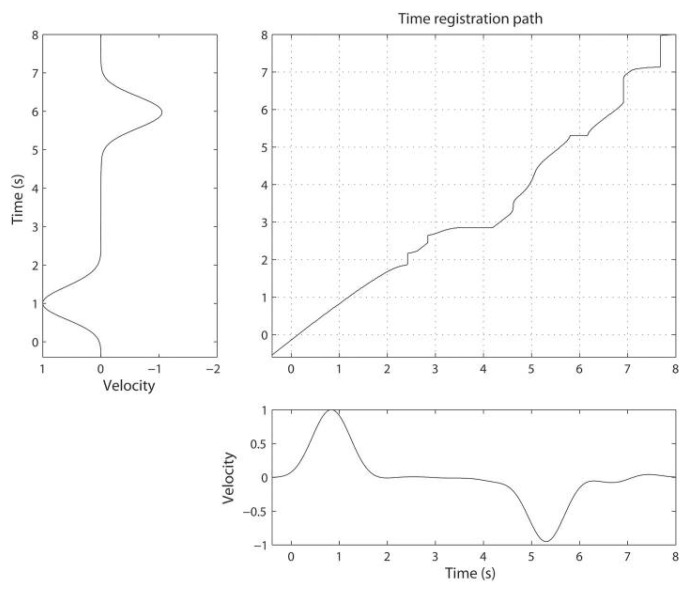
An illustration of accumulated cost matrix and associated optimal warping path when using DTW to align the two time-series in the picture.

**Figure 2 sensors-19-05192-f002:**
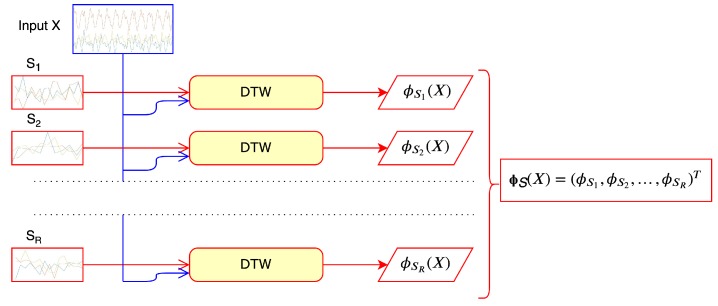
Illustration of the parallel computation of the components of the vector $\Phi_{S}\left(X\right)$. Time-series in the figure are assumed multivariate.

**Figure 3 sensors-19-05192-f003:**
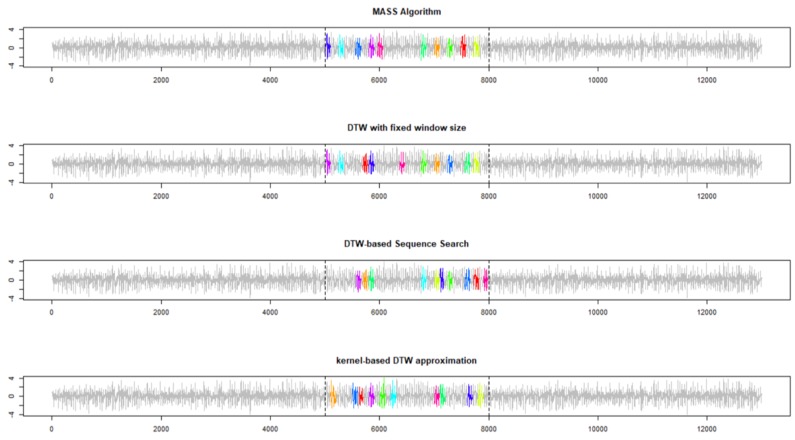
The first 10 best-matching subsequences were identified by four different algorithms on the Experiment 1. The reference pattern consists of 100 acceleration data points collected when a Sony AIBO robot was walking on a carpet. The data stream consists of 5000 data points when the same dog was walking on cement, then followed by 3000 walking on a carpet and then again on cement.

**Figure 4 sensors-19-05192-f004:**
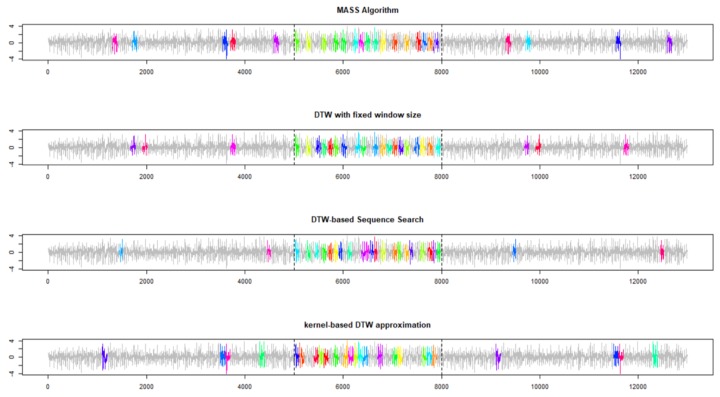
First 25 best matching subsequences identified by four different algorithms on the Experiment 1. Reference pattern consists of 100 acceleration data points collected when a Sony AIBO robot was walking on a carpet. The data stream consists of 5000 data points when the same dog was walking on cement, followed by 3000 walking on a carpet and then again on cement.

**Figure 5 sensors-19-05192-f005:**
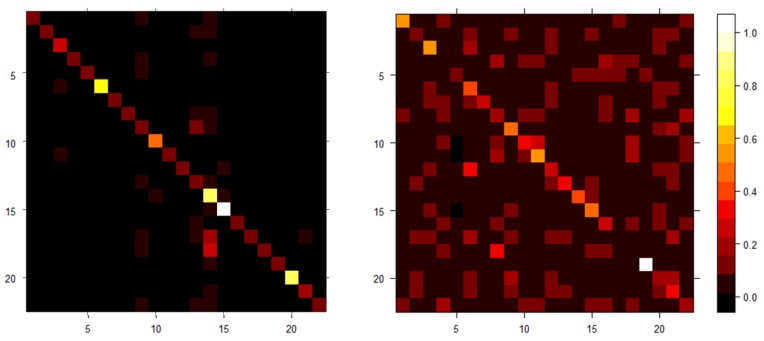
Distance between each pair of reference and data stream, respectively, for DTW-based sequence search (on the **left**) and kernel-based DTW approximation (on the **right**). The brighter the colour the lower the distance.

**Table 1 sensors-19-05192-t001:** DTW-based subsequence search vs. kernel-based DTW approximation for subsequence search.

	Coherent Visually	
Exercise	DTW-Based Subsequence Search	Kernel-Based DTW Approximation
Flexo-extension of the knee, sit-down position. Five repetitions planned.	Yes	Yes
	Yes	Yes
	Yes	Yes
	No	Yes
	No	Yes
Light squat, stand-up position. Five repetitions planned.	Yes	No
	Yes	No
	Yes	Yes
	Yes	-
	Yes	-
Back extension of the legs, stand-up position. Five repetitions planned.	Yes	Yes
	Yes	Yes
	Yes	Yes
	No	Yes
	No	Yes
Rotate the torso, sit-down position. Five repetitions planned.	Yes	Yes
	Yes	Yes
	Yes	-
	Yes	-
	No	-
Raise and lower the arms, sit-down position. Ten repetitions planned.	Yes	Yes
	Yes	Yes
	Yes	Yes
	Yes	Yes
	Yes	Yes
	No	Yes
	Yes	Yes
	No	Yes
	Yes	-
	Yes	-

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
