# Peer review of "Efficient Kernel-Based Subsequence Search for Enabling Health Monitoring Services in IoT-Based Home Setting"

_sensors, 2019, doi:10.3390/s19235192_

Round 1
Reviewer 1 Report
The paper presents an approach to learn a kernel to approximate DTW for time-series analysis. The idea sounds reasonable and interesting. The experimental results are convincing. However, I have the following concerns regarding the current paper.
(1) In Introduction, more related work should be introduced. The motivation of this paper should be highlighted.
(2) One of the advantages of the proposed approach is the low computational cost. Can you compare the running time regarding the proposed approach and the DWT-based method in Experiment 2 and 3?
(3) It seems that the approach uses RBF kernel and then optimize the parameter \gamma in RBF by BO. Is it true? I wonder if it is possible to adopt a linear kernel instead of RBF? If so, no parameters need to be optimized.
(4) Besides the "1 core 1.7 HZ", what are the experimental environments, including os, software and other hardware? Do the authors use parallel computation?
(5) How to set the parameters for Algorithm 3 in the experiments? Has Algorithm 3 been validated in the paper?
(6) Experiment 3 should be discussed and analyzed in detail.
(7) The experiments are all conducted on small-scale datasets. Can the proposed scheme be used for large-scale datasets?
Author Response
(1) In Introduction, more related work should be introduced. The motivation of this paper should be highlighted.
We have added two more references – as suggested by the second reviewer – to further highlight how widely DTW is adopted and computational issue can be relevant. With respect to this point, most of the relevant research works and open issues are reported in the Background (Section 2): in introduction we have added a clear reference to this.
(2) One of the advantages of the proposed approach is the low computational cost. Can you compare the running time regarding the proposed approach and the DWT-based method in Experiment 2 and 3?
Thank you for the comment. We have reported the running time in the experimental results of each one of the three experiments.
(3) It seems that the approach uses RBF kernel and then optimize the parameter \gamma in RBF by BO. Is it true? I wonder if it is possible to adopt a linear kernel instead of RBF? If so, no parameters need to be optimized.
Thank you for this comment. Indeed, we decided to use RBF in order to exploit the possibility to deal with non-linear embedding and to increase the capabilities of approximation of the approach. Naturally, this leads to need for optimizing the length-scale hyperparameter of the kernel. We have added a sentence to clarify and motivate our choice.
(4) Besides the "1 core 1.7 HZ", what are the experimental environments, including os, software and other hardware? Do the authors use parallel computation?
We have added a new sub-section related to the computational setting, including hardware, OS and software tools. (Furthermore, the information about hardware have corrected because a wrong description was reported).
(5) How to set the parameters for Algorithm 3 in the experiments? Has Algorithm 3 been validated in the paper?
The computational complexity analysis reported in sub-section 3.2. “Learning a kernel for subsequence search” drives the choice of the technical parameters of the Algorithm 3. We have added the following sentence, at the end of sub-section 3.3., clearly stating it. Algorithm 3 is the extension of the approach to the case of multi-references and it was specifically validated through Experiment 2. Also this point is now clearly stated in the text.
(6) Experiment 3 should be discussed and analyzed in detail.
Thank you for this comment. Indeed, the description of this experiment was not so clear – as also reported by the other reviewer. We have improved the description for this experiment.
(7) The experiments are all conducted on small-scale datasets. Can the proposed scheme be used for large-scale datasets?
Thank you for this comment. Yes, the approach can be applied to large-scale dataset and the adoption of the parallelization schema proposed in Figure 2 can ensure a good scalability. We have added a sentence in the Conclusions section to make this finding evident.
Reviewer 2 Report
The paper presents a DTW kernel-based method for subsequence search in long time series. The aim is to reduce the complexity of DTW computation by using kernel approximations over a randomly chosen basis of short sequences. In general, the paper is well-written, but improvements can still be made regarding the evaluation of proposed method during the experiments, connections with other methods that could benefit from the proposed approach and choice of the random basis.
1. The main concern regarding the approach is the choice of a random basis. The authors mention that this aspect will be part of future research. However, this reviewer considers that the choice of the basis is extremely important. In this sense, the authors might find useful:
Ye, L., & Keogh, E.J. (2009). Time series shapelets: a new primitive for data mining. KDD
2. DTW has been used in many algorithms for clustering. The authors mention some of these algorithms (fuzzy clustering [10], SVM [9]), but other algorithms would also benefit from the kernel-based DTW method introduced in this paper and its speed-up property. For example:
EM-DTW (Radoi, A.; Burileanu, C. Retrieval of Similar Evolution Patterns from Satellite Image Time Series. Appl. Sci. 2018, 8, 2435.)
NN-DTW (F. Petitjean, G. Forestier, G. I. Webb, A. E. Nicholson, Y. Chen and E. Keogh, "Dynamic Time Warping Averaging of Time Series Allows Faster and More Accurate Classification," 2014 IEEE International Conference on Data Mining, Shenzhen, 2014, pp. 470-479.)
3. Some more details should be added in subsection 3.3. For example, some explanations regarding relation (14).
4. In equation (12), \bar{\rho} should be between 0 and 1.
5. In Algorithm 3, line 5 should be included in for loops.
6. Page 12, line 291: 8 instead of 7.
7. Table 1 is not very relevant. Instead, the author could pick random subsequences and perform a significant number of tests and then report percentages for the hit and miss rates. In this sense, in the other two experiments, the authors should report similar performance measures.
8. Some minor corrections: "do" instead of "does" (line 116), "hichever", additional "a" (line 188).
Author Response
The paper presents a DTW kernel-based method for subsequence search in long time series. The aim is to reduce the complexity of DTW computation by using kernel approximations over a randomly chosen basis of short sequences. In general, the paper is well-written, but improvements can still be made regarding the evaluation of proposed method during the experiments, connections with other methods that could benefit from the proposed approach and choice of the random basis.
(1) The main concern regarding the approach is the choice of a random basis. The authors mention that this aspect will be part of future research. However, this reviewer considers that the choice of the basis is extremely important. In this sense, the authors might find useful:
Ye, L., & Keogh, E.J. (2009). Time series shapelets: a new primitive for data mining. KDD
Thank you so much for the suggested reference. We have added it in the conclusions as a relevant starting point for future works
(2) DTW has been used in many algorithms for clustering. The authors mention some of these algorithms (fuzzy clustering [10], SVM [9]), but other algorithms would also benefit from the kernel-based DTW method introduced in this paper and its speed-up property. For example:
EM-DTW (Radoi, A.; Burileanu, C. Retrieval of Similar Evolution Patterns from Satellite Image Time Series. Appl. Sci. 2018, 8, 2435.)
NN-DTW (F. Petitjean, G. Forestier, G. I. Webb, A. E. Nicholson, Y. Chen and E. Keogh, "Dynamic Time Warping Averaging of Time Series Allows Faster and More Accurate Classification," 2014 IEEE International Conference on Data Mining, Shenzhen, 2014, pp. 470-479.)
Thank you so much for the suggested references, they are really interesting and further improve the quality of the Introduction section.
Some more details should be added in subsection 3.3. For example, some explanations regarding relation (14).
We added a more detailed description of the equation (14).
In equation (12), \bar{\rho} should be between 0 and 1. 5. In Algorithm 3, line 5 should be included in for loops.
Thank you for the comment: modification has been applied.
Page 12, line 291: 8 instead of 7.
Thank you for the comment: modification has been applied.
Table 1 is not very relevant. Instead, the author could pick random subsequences and perform a significant number of tests and then report percentages for the hit and miss rates. In this sense, in the other two experiments, the authors should report similar performance measures.
We do not agree with the proposed validation schema. We are interested in identifying specific reference patterns that are “meaningful”, not random extracted portions from a stream. Probably the description of the experiment 3 was so poor to lead to a misunderstanding of the validation goal. The poor quality of the description was also remarked by the other reviewer. We have improved this section and we are now confident that the goal of the validation, as well as the corresponding procedure are clearer.
Some minor corrections: "do" instead of "does" (line 116), "hichever", additional "a" (line 188).
Thank you for the comment: modification has been applied.
Round 2
Reviewer 1 Report
Since all my concerns have been well addressed, I think the paper can be accepted now.
Some minor errors: L357, "Table ??", "he subsequences".
Author Response
Thank you so much for your comment. Changes have been applied.
Reviewer 2 Report
All my comments have been adequately addressed. The manuscript is acceptable in this form regarding theoretical presentation, experimental part, results. Some minor comments:
(1) No space after 'case' (line 3), no punctuation sign after 'vector' (line 9), additional 'a' (line 11), additional 'dataset' (line 47), Table ?? (line 357), 'he subsequences' (line 357), additional 'the' (line 368).
(2) The authors should re-check the notations used in Subsection 3.3 and Algorithm 3 (e.g., X_i, X_j, Y_j, i, j, m, n). Maybe using j to index X series and k for Y series instead of using same j index for both (i.e., j = 1...n and k = 1...m). Line 4 in Algorithm 3 should be 'for j = 1:n'.
Author Response

(The authors gave the same response as above.)
